# Fractionated Volumetric Modulated Arc Therapy (FVMAT) for Oligometastatic Brain Tumor

**Chi-Yuan Yeh ***, **Peng-An Lai, Fang-Hui Liu and Chin-Chiao He**

Department of Radiation Oncology, Tungs' Taichung Metroharbor Hospital, No. 699 Sec. 8 Taiwan Boulevard, Wuqi District, Taichung 43503, Taiwan
* Correspondence: peteryeh46@gmail.com; Tel.: +886-04-26581919 (ext. 4385)

**Simple Summary:** Neurocognitive deficit is often encountered in cancer patients with brain metastasis receiving whole brain radiotherapy (WBRT); this is due primarily to the radiation-induced toxicity to the hippocampal region of the brain, which is vital for verbal learning and memory function. Volumetric modulated arc therapy (VMAT) is a new type of radiotherapy that has a sparing effect on normal brain tissues while maintaining an adequate radiation dose to the malignant brain tumors. We aim to promote this technique for clinical application, working towards the goal of improving life quality for these patients.

**Abstract:** Intracranial metastasis is very common in adult cancer patients with an overall incidence of approximately 10–40%. The most common primary tumors responsible for this in adults are lung and breast cancer. Brain metastasis signifies a grave prognosis, with a median survival of 6 to 12 months. They are traditionally managed with palliative care and whole brain radiotherapy (WBRT). WBRT was an effective method to control brain metastases, decreasing corticosteroid use to control tumor-associated edema, and potentially improving overall survival; however, WBRT was found to be associated with a serious neurocognitive degeneration, this adverse effect (AE) follows a biphasic pattern beginning with a transient decline in mental functioning at around 4 months post-treatment, slowly leading to an irreversible neurologic impairment from months to years later. Evidence supports that WBRT can cause radiation injury to the hippocampus, which in turn will lead to a decline in neurocognitive function (NCF). Volumetric modulated arc therapy (VMAT) is a relatively new type of image-guided radiotherapy that treats multiple brain metastasis simultaneously and efficiently with less neurocognitive sequelae. Eighteen cancer patients with limited (≤5 brain tumors) or oligometastatic brain tumor were treated with a spatially fractionated VMAT technique for a total dose of 30 Gy in 10 fractions, the patients tolerated the VMAT treatment with no radiation-induced neurologic toxicities after a mean follow-up of 1 year. Local control rate was 84%, and the median survival for these 19 patients was 11.3 months (range: 9.1–22.4 months). In conclusion, the VMAT is a suitable technique that is a safe and effective treatment for brain oligometastases.

**Keywords:** radiotherapy; whole brain radiotherapy (WBRT); volumetric modulated arc therapy (VMAT); oligometastasis (OM)

## 1. Introduction

Oligometastases (OM) is defined as a special cancer state wherein the primary tumor is controlled, and the patient has a limited 1–5 distant metastatic tumor limited to a specific organ; this is a special biologic state wherein cancer has a restricted metastatic capacity.

This concept was first proposed by Hellman and Weichselbaum [1] in 1995 as an intermediate state in which aggressive local therapy could achieve good tumor control and maybe curative in these patients who by strict definition are already terminal stage 4 patients. They postulated that in the natural clinical evolution of a cancer, a progression

of malignancy is observed as evidenced by the increasing metastatic capacity, leading to systemic dissemination, and, ultimately, to the death of the patient if this condition is not controlled.

Huang et al. [2] showed that the prognosis is inversely related to the number of metastatic lesions; their review confirmed that 4 is a critical number for outcome. Long-term survival was deemed possible by a study that treated 121 OM patients with stereotactic body radiotherapy (SBRT) and showed a 6-year overall survival (OS) of 20%, and 47% OS for selected breast cancer patients with OM [2]. The key issue for OM patients is how to identify the most suitable population that will benefit from curative and aggressive local treatment.

The Radiation Therapy Oncology Group (RTOG) 91-04 [3] developed three prognostic classes for brain metastases using recursive partitioning analysis (RPA) of a large database; RPA class 1 and class 2 were identified by the study as benefiting the most from aggressive local treatment. Class 1 patients have a Karnofsky performance status (KPS) >70, <65 years old with controlled primary and no extracranial metastasis, while class 2 have a KPS >70 with uncontrolled primary or extracranial metastasis.

Weichselbaum [4] proposed that noninvasive regional treatment such as chemotherapy, stereotactic radiotherapy, radiofrequency ablation, and focused ultrasound can potentially cure and improve the cancer patient's survival, since these oligometastatic tumor cells were more sensitive while tumor clones with high metastatic potential were found to be more resistant to cytotoxic treatment.

The incidence of brain metastases was estimated to be from 8.3 to 14.3 per 100,000 population [5]. An increase in awareness and early diagnosis of brain OM was due to the widespread availability of magnetic resonance imaging (MRI) in the clinics. The occurrence of brain metastasis is clinically challenging as it is associated with poor prognosis, neurological dysfunction, and reduced quality of life, and confers a grave prognosis, with an estimated median survival of 1 year or less [5,6]. However, new systemic targeted therapy has been shown to improve the survival for these patients. Osimertinib is a relatively newer type of tyrosine kinase inhibitor that has been shown to result in a significantly longer progression free survival in nonsmall cell lung carcinoma with epidermal growth factor receptor (EGFR) with T790M mutation. The T790M cohort OCEAN study has shown that osimertinib at a dose of 80 mg once daily for radiotherapy-naive brain metastasis has resulted in a median overall survival of 25.2 months [7].

It is considered an end-stage disease; up to 30–40% of cancer patients develop brain metastases at some point during the course of their disease, the three most common primary cancer sites [5,6] are metastasis from the lungs (45–50%), breast (10–30%), and melanomas (5–20%).

Whole brain radiotherapy (WBRT) has been the mainstay of treatment for multiple brain metastasis since the 1950s due to its effectivity in relieving neurologic symptoms, widespread availability, and short treatment time. Chao et al. [8] first described treating brain metastasis using WBRT for a cumulative dose of 2000 roentgen to 4100 roentgens in 1954, the mean survival for 14 patients who responded to WBRT was 8.2 months, while it was 4.6 months for 12 poor responder patients. Geber et al. [9] in an RTOG clinical trial treated 1830 patients with WBRT with different radiation dose fractionation; their study concluded that a total dose schedule of 20 Gy in 5 fractions or 30 Gy in 10 fractions was equivalent to 40 Gy in terms of palliation effect and neurological toxicity.

WBRT has been the standard treatment for multiple brain metastasis, with 60% showing complete or partial response and 50% good palliative response. However, recent studies have demonstrated cognitive decline; these studies showed that 35% to 52% of patients had a 3–standard deviation (SD) drop on cognitive testing 3 to 6 months after WBRT. WBRT alternatives included the Japanese Leksell Gamma knife (JLGK) trial 0901 showing promising results, this is multi-institutional prospective observational trial where 1194 patients with 1 to 10 brain OM were treated with radiosurgery alone. Another attractive alternative is hippocampal avoidance WBRT(HA-WBRT), this technique uses a modern intensity modulated

radiotherapy technique (IMRT) that conformally avoids irradiating the hippocampus while delivering the therapeutic radiation dose to the whole-brain parenchyma, this technique has been shown to preserve memory and neurocognitive function [10]. Indeed, the recently conducted phase III NRG oncology CC001 trial for 518 patients comparing HA-WBRT plus memantine with WBRT plus memantine showed a lower cognitive failure for HA-WBRT when compared with WBRT; the adjusted hazard ratio was 0.74 with a *p* value of 0.01 [11].

Volumetric modulated arc therapy (VMAT) is a new IMRT technique wherein radiotherapy is delivered using a continuous arc motion of the linear accelerator gantry around the patient, accompanied with simultaneous modulation of the multileaf collimator (MLC) position, gantry rotation speed, and dose rate output per angle [12].

Clark et al. [13], in a feasibility study, concluded that a single-isocenter VMAT can be used to deliver conformity equivalent to the multiple isocenter VMAT radiosurgery technique; this technique is extremely efficient, requiring less than one-half of treatment time required for multiple targets using multiple isocenters. Ohira et al. [14] recently treated 23 brain OMs (1–4 brain metastasis) using a hyperArc VMAT approach in a single stereotactic radiosurgery (SRS) dose of 20 to 24 Gy, this technique uses a single-isocenter, non-coplanar flattening filter free beam arrangement. They concluded that the hyperarc technique is much superior to the conventional VMAT technique in terms of higher dose conformity and rapid dose falloff near the periphery of the target area.

Taiwan's health care insurance only covers the cost for one-time treatment of a single brain lesion SRS either using a gamma knife or a linear accelerator; the policy for treating multiple brain lesions is unclear, and since the hyperArc VMAT is not available, WBRT has been historically used to treat brain metastasis patients, with good palliative results but accompanied by evident neurologic sequela after long-term follow-up.

We retrospectively reviewed 18 patients of brain OM tumor who were treated simultaneously with the fractionated single isocenter VMAT technique to assess its clinical benefit.

## 2. Materials and Retrospective Review

A retrospective review was made for 18 patients with brain oligometastases who were referred to the department of radiation oncology from 1 January 2018 to 1 February 2021 for treatment. The patient characteristics are shown in Table 1. The eligibility criteria for VMAT treatment are the following: the patient's performance status according to eastern cooperative oncology group (ECOG) [15] should be grade 0 (fully active with no restriction), grade 1 (ambulatory and able to carry light work), and grade 2 (up and about >50% of waking hours); ECOG grade 3 (confined to bed or chair >50% of waking hours) and grade 4 (totally confined to bed) patients were excluded. All 18 patients had their primary tumor verified pathologically by either biopsy or surgical resection, all patients after curative treatment had their primary tumor under control with no recurrence, but during follow-up were shown to have brain oligometastases proven with a gadoterate meglumine contrast-enhanced magnetic resonance imaging (MRI) of the brain.

A total of 61 brain oligometastatic lesions in 18 patients were treated using the fractionated volumetric modulated arc technique (FVMAT), the mean number of brain OM per patient is 3.4 (range: 1–6 lesions) with a mean volume of 65.0 cm$^3$. All patients had not been previously treated with WBRT. A Varian TrueBeam STx linear accelerator (Varian Medical Systems, Inc., Palo Alto, CA, USA) equipped with an HD120 multileaf collimator (MLC) was used to simultaneously treat all brain oligometastatic tumors with two arcs (Figure 1) at 600 monitor units (MU) per minute, one MU being equivalent to 0.01 Gray (Gy). The central high resolution leaf width of the HD120 MLC was 2.5 mm.

**Table 1.** Patient characteristics.

| Characteristics | Number |
|---|---|
| Male | 12 |
| Female | 6 |
| Age | |
| Mean | 61.6 |
| Range | 23–74 |
| Primary tumor | |
| Lung | 10 |
| Breast | 2 |
| Tongue | 1 |
| Prostate | 2 |
| Urinary bladder | 1 |
| Gastric | 1 |
| Ewing's sarcoma | 1 |
| Number of brain OM lesions | |
| 1 | 1 |
| 2 | 4 |
| 3 | 6 |
| 4 | 3 |
| 5 | 2 |
| 6 | 2 |
| OM total tumor volume | |
| Mean | 65.0 cm$^3$ |
| Range: | 2.1–173 cm$^3$ |
| ECOG | |
| 0 | 5 |
| 1 | 11 |
| 2 | 2 |

The patient's head was immobilized via a rigid thermoplastic head frame docked to a Qfix intracranial baseplate with the appropriate head support. Then, a 3 mm slice thickness computed tomography (CT) simulation of the patient's brain region was first performed with a Somatom definition AS 16-slice scanner (Siemens Healthineers, Erlangen, Germany) before treatment. The CT scan images were then registered and fused with T1-weight MRI images of the individual patient. The clinical target volume (CTV) and surrounding organs-at-risk (OAR) were delineated based on the fused CT–MRI image using the tumor contouring function of the Varian Eclipse radiotherapy treatment planning system, version 13.6 (Varian Medical Systems, Inc., Palo Alto, CA, USA). A cumulative dose of 30 Gy in 10 fractions was prescribed to the planning target volume (PTV). The PTV encompassed the CTV, regions of microscopic potential spread, and a circumferential margin of 3 mm is added to the CTV to account for setup errors and potential patient motion, at least 95% of the PTV should be covered by 95% of the prescribed dose to ensure adequate dose coverage [16]. The anisotropic analytical algorithm (AAA v13.0.26) of the Eclipse treatment planning system (Varian Medical Systems, Inc.) was adopted for dose calculation with a calculation grid of 2 mm × 2 mm. The calculated 95% isodose line was prescribed to cover 95% of the PTV volume. The dose constraints for organs-at-risk (OAR) used for our VMAT treatment (Table 2) were defined according to the RTOG 0933 protocol [17]. These 18 patients were then followed up every 3 months for at least 1 year for physical check-up, lab tests such as complete blood count, liver function test, renal function test, fasting blood glucose, tumor markers, and urine test, with brain MRI being performed 6 months later to assess tumor response, and development of new metastatic lesions. The patients were then treated with further chemotherapy or hospice care if there were any new metastatic brain lesions, for partial response of brain tumor to radiation therapy. No re-irradiation of new brain lesion was done in our hospital as per treatment protocol. Their performance status to evaluate the neurotoxicity effect of VMAT treatment based on the basic Activities

of Daily Living (Table 3) score [18] was assessed every 3 months, an activities of daily living (ADL) score of 6 and above connotes a patient totally dependent on care giver.

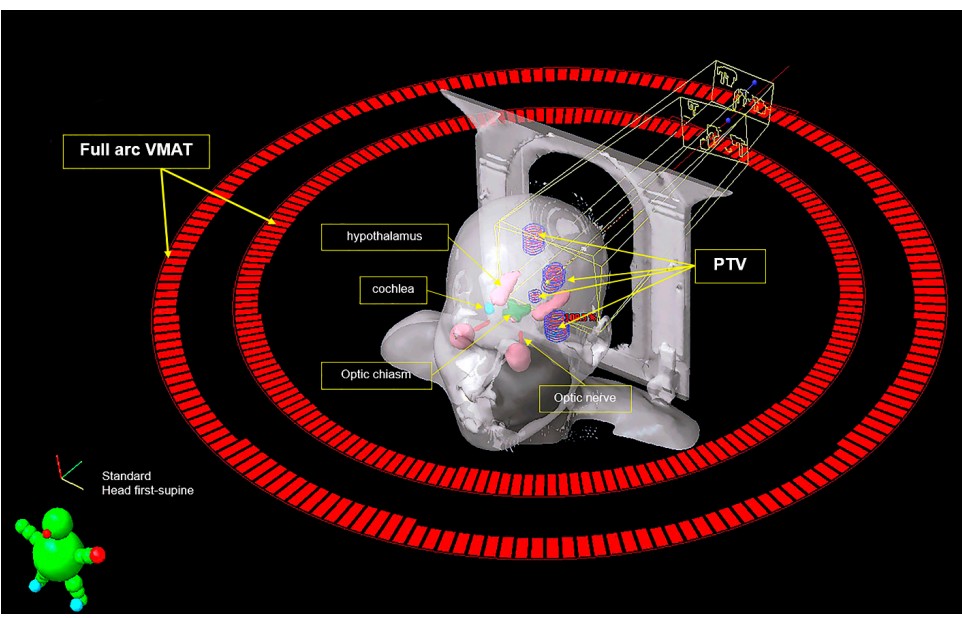

**Figure 1.** Fractionated VMAT treatment for brain oligometastases.

**Table 2.** Dose constraint for normal organs-at-risk (OAR).

| Organs | D100% [1] | Dmax [2] |
|---|---|---|
| hippocampus | ≤10 Gy | ≤17.0 Gy |
| optic chiasma | | ≤37.5 Gy |
| optic nerve | | ≤37.5 Gy |
| eyes | | 7 Gy |
| lenses | | 5 Gy |

[1] dose to 100% of the hippocampal volume. [2] maximum dose.

**Table 3.** Scale of basic activities of daily living (ADL).

| | Activity | Description | Score |
|---|---|---|---|
| 1. | Hygiene | Autonomous | 0 |
| | | Partial assistance for one part of body | 1 |
| | | Assistance for several parts of the body or toileting impossible | 2 |
| 2. | Dressing | Autonomous | 0 |
| | | Dresses but needs assistance with shoes Needs assistance in choosing clothing, getting dressed, and remains partially dressed | 1 |
| | | Completely undressed | 2 |
| 3. | Toileting | Autonomous | 0 |
| | | Needs to be accompanied; needs assistance | 1 |
| | | Does not go to the toilet; does not use the toilet or urinal | 2 |
| 4. | Locomotion | Autonomous | 0 |
| | | Needs assistance | 1 |
| | | Bedridden | 2 |

**Table 3.** *Cont.*

| | Activity | Description | Score |
|---|---|---|---|
| 5. | Continence | Continent | 0 |
| | | Occasional incontinence | 1 |
| | | Permanent incontinence | 2 |
| 6. | Meals | Autonomous | 0 |
| | | Needs assistance to cut meat or peel fruit | 1 |
| | | Total assistance or artificial feeding | 2 |

ADL score of >6 = sign of dependence.

## 3. Results

The median survival for the 18 patients was 11.3 months (range: 9.1–22.4 months); follow-up MRI scan 6 months after VMAT therapy showed good local control of brain oligometastases with no regrowth or new tumor growth; all 18 patients did not suffer any chronic neurologic sequelae or cognitive decay during monthly follow-up; all 18 patients were able to maintain an ECOG 0 to 2 performance status during follow-up; short-term grade 1 alopecia, nausea, anorexia, heavy-sensation of head was reported by all patients, but oral dexamethasone relieved these symptoms by alleviating increased intracranial pressure caused by brain irradiation. Table 4 shows that all patients' ADL were maintained at an acceptable score of <6 six months after FVMAT treatment. Only 1 patient with an ADL score of 5 suffered a deterioration of physical condition 13 months later due to systemic disease and was referred to hospice care.

**Table 4.** Total ADL score of 18 patients evaluated 6 months after FVMAT.

| Total ADL Score | Patients | % |
|---|---|---|
| 1 | 0 | 0.0% |
| 2 | 3 | 16.7% |
| 3 | 9 | 50.0% |
| 4 | 5 | 27.8% |
| 5 | 1 | 5.6% |
| $\geqq 6$ | 0 | 0.0% |
| | 18 | 100.0% |

Figure 1 depicts a typical VMAT setup using a coplanar double full arc to irradiate 4 brain oligometastatic tumors simultaneously for a cumulative dose of 30 Gy in 10 fractions in 12 days. This patient is a 70-year-old heavy smoker male patient who was diagnosed with squamous cell carcinoma of the right lower lung cancer clinically staged as T4N3M0, stage IIIc, complicated with chronic obstructive pulmonary disease in 30 March 2020; the chief complaint was persistent productive cough with blood-tinged sputum for 4 months associated with poor appetite. Adjuvant chemotherapy with paclitaxel at 45 mg per m$^2$ body surface area (BSA), cisplatin 25 mg/m$^2$ BSA for 6 course was given concurrently with 60 Gy irradiation (10 April 2020–25 May 2020) to left lower lung cancer and mediastinal lymph node. The patient tolerated the treatment well with an ECOG 1 performance status. A chest CT scan on 17 February 2021 showed the primary lung tumor well under control with no local recurrence; however, the patient began to have unexplained nausea, vomiting with heavy sensation of the head, and a contrast-enhanced brain MRI on 2 April 2021 revealed four peripherally ring-enhanced tumors 0.4 to 2 cm in size located in the cerebellar vermis, left temporal lobe, and bilateral occipital lobe of the brain. The patient tolerated the VMAT procedure well, maintaining an ECOG 1 performance status, side effects were partial temporary alopecia, a heavy sensation of the head that was relieved by oral steroid

medication, and the patient was still under follow-up 12 months after VMAT with no overt memory lost, visual, speech, or neurocognitive impairment.

Table 5 shows the volume (cm³) and dose received for each OAR that are critical for the normal neurocognitive function of the patients; the dose for various critical OARs were within the recommended tolerance dose listed in Table 2. The CTV and PTV dose were within the prescribed dose of 30 Gy with <5% variation.

**Table 5.** Average dose to OARs for 18 brain oligometastatic patients treated with FVMAT.

| Structure | Volume(cm³) | Min (Gy) [1] | Max (Gy) [1] | Mean (Gy) [1] |
|---|---|---|---|---|
| brain | 1298.7 | 0.7 | 32.2 | 9.1 |
| brainstem | 31.1 | 0.8 | 11.9 | 4.0 |
| left eye | 9.4 | 0.7 | 6.0 | 1.8 |
| right eye | 9.4 | 0.7 | 5.0 | 1.6 |
| left optic nerve | 1.0 | 1.7 | 3.3 | 2.3 |
| right optic nerve | 0.7 | 1.5 | 3.9 | 2.4 |
| left lens | 0.3 | 1.0 | 1.6 | 1.2 |
| right lens | 0.3 | 0.9 | 1.6 | 1.2 |
| chiasm | 3.3 | 2.7 | 6.2 | 3.8 |
| left hippocampus | 3.7 | 3.3 | 8.2 | 5.2 |
| right hippocampus | 3.5 | 3.9 | 9.7 | 5.7 |
| right cochlea | 0.7 | 2.6 | 5.2 | 3.7 |
| left cochlea | 0.7 | 2.2 | 5.6 | 3.2 |
| pituitary gland | 0.4 | 2.8 | 3.9 | 3.3 |
| CTV | 65.0 | 27.9 | 32.2 | 31.2 |
| PTV | 103.2 | 20.0 | 32.4 | 30.8 |

[1] minimum, maximum, mean dose in Gray.

## 4. Discussion

Multileaf collimator (MLC) was introduced in the 1980s to make intensity modulated radiation therapy or VMAT treatment a reality. These techniques are now an integral role in the curative treatment of cancers worldwide. MLC is made of tungsten leaves that move independently under computer control; this movement forms a radiation field with varying shapes and intensity while the linear accelerator rotates around the patient, thus enabling a radiation field that conforms to the complicated shape of the tumor 3-dimensionally [19]. The mean dose reduction to the normal tissue attributable to the HD120 MLC was between 1 and 4% for the 3-dimentional conformal radiotherapy and between 2 and 6% for the intensity modulated radiotherapy (IMRT) technique, an improvement in conformity index with a target volume for all treatment techniques, and the best PTV coverage was noted for the HD-MLC system [20].

Conventional 2-dimensional grid techniques use only one field with a physical grid; this method does not have the 3-dimensional conformal dose distribution in the current multiple field IMRT or VMAT technique. VMAT would be an ideal approach to deliver a tightly conformal high dose within the PTV with a shorter treatment time when compared with conventional radiotherapy. Hsu et al. [21] conducted a feasibility study of WBRT with hippocampal avoidance and simultaneous integrated boost (SIB) for one to three brain metastases with the VMAT technique; ten patients with 18 brain metastasis were treated with WBRT for a total dose of 32.25 Gy in 15 fractions, SIB technique to individual brain tumor were given a total dose of 63 Gy for >2 cm lesions and 70.8 Gy for <2 cm lesions, and each brain tumor was treated using a different VMAT plan and a unique isocenter. Their result showed that VMAT was able to achieve adequate whole brain coverage with conformal hippocampal avoidance and radiation quality dose distributions for one to three brain metastases; the mean hippocampal dose for this study was 5.23 ± 0.39 Gy, which is below the mean dose constraint of 6 Gy.

The hippocampus is a paired brain structure that is situated laterally to the temporal horn of the lateral ventricles forming the limbic system, it is embedded in the temporal lobe of the brain, and its function is mainly centered on learning, consolidation, and retrieval of information and long-term memories [21]. Studies have shown that a biologically equivalent dose greater than 7.3 Gy (equivalent dose in 2-Gy fractions) to the hippocampus will result in long-term neurocognitive function impairment; meaningful reduction in the hippocampal dose while maintaining acceptable tumor control probability has been established with VMAT therapy [22]. The RTOG 0933 was a single-arm phase II study that accrued 113 adult patients from 2011 to 2012; WBRT for 30 Gy in 10 fraction was prescribed for these patients with multiple brain metastases. They compared the historical control of patients treated with WBRT without hippocampal avoidance and the present WBRT protocol with hippocampal avoidance. The mean relative decline in the Hopkins Verbal Learning Test–Revised Delayed Recall (HVLT-R DR) from baseline to 4 months was 7.0%, significantly lower in comparison with the historical control ($p < 0.001$). No decline in QOL scores was observed. Two grade 3 toxicities and no grade 4 to 5 toxicities were seen for these patients. The median survival was 6.8 months. Avoidance of the hippocampus during WBRT is associated with preservation of memory and quality of life as compared with historical series [23]. Our FVMAT patient series did not show any sign of brain necrosis with MRI scan or any sign of neurologic decline; as seen in Table 4, the ADL score after 6 months follow-up was within normal limits, this is due to the fact the FVMAT technique has the advantage of avoiding direct irradiation of the hippocampus while treating simultaneously all brain OMs.

Andrevska [24] in a review noted that the long treatment times incurred by stereotactic radiosurgery (SRS) techniques prompted the need for an alternative technique that has shorter treatment times, while still maintaining a highly conformal radiation field around the brain OM lesion. SRS treatment time can vary from 30 min for a single lesion to several hours for multiple lesions, with treatment time increasing with the number of lesions; treatment time is further lengthened when using cone-based SRS technique. Four of the SRS studies in this review reported incidence of brain necrosis as a late effect commonly experienced due to the high doses delivered [24].

Fractionated VMAT, on the other hand, has been proven to be a safer alternative; a study involving 115 patients with a reported 1 to 10 brain metastases treated with FVMAT (40 Gy to 50 Gy in 12 to 15 fractions) reported no significant neurologic toxicities and had achieved a 79–81% local control at 12 months, the median overall survival ranged from 9 to 10 months. Our study revealed a similar experience with our local control rate being 84% and median survival being 11.3 months.

Furutani et al. [25] treated 67 patients with 601 brain metastases with hypofractionated image-guided multifocal irradiation using VMAT (HFIGMI–VMAT) from 2012 to 2016, the prescribed dose was 50 Gy to a 95% volume of the PTV in 10 fractions, the median number of brain metastases per patient was 5 (range: 1–73), and the median volume of the brain lesion was 1.0 cm$^3$ (range: 0.1–14.1 cm$^3$); the median survival time was 18.7 months. The local control rate was 98.4% and 95.3% at 6 and 12 months, respectively. Brain tumor size has a direct correlation with local control; the local control rates at 12 months were 100% for tumors of diameter $\leq 2$ cm, which was significantly better than the 83.3% for tumor diameters of >2 cm ($p < 0.0001$). Late radiation necrosis occurred in ten patients (14.9%), three (4.5%) of whom required corticosteroid therapy. This study showed a better local control than our series but reported a higher incidence of radiation-induced brain necrosis, while our patients have not experienced this complication; this is probably due to the higher radiation dose of 50 Gy in Furutani's study [25], and this is the reason we limited our dose to 30 Gy.

By 2014, the shift from treating a single lesion to multiple brain metastases gathered momentum as new technologies made SRS possible without incurring severe neurological toxicity by sparing healthy brain tissue from the toxic levels of radiation. This new treatment has resulted in longer survival and improved quality of life for the patients [26]. For brain

OM, SRS uses one fraction of radiation with a single isocenter per target, the treatment can take up to several hours, delivery is prolonged by the need for couch and gantry movement, resulting in lengthy treatment times [26]; our FVMAT technique is superior to single fraction SRS because of a faster treatment time of less than 5 min, treating all brain OM simultaneously in one isocenter with no couch movement, and the mask-based treatment for FVMAT enables hypofractionation to reduce radiation-induced neurotoxicity with comparable tumor local control. The total number, size, and relative proximity between each metastasis has a more important bearing on radiation treatment plan quality than the radiation technique themselves. This is not a key issue for us because the Eclipse treatment planning system can optimize the FVMAT radiation dose distribution for all brain OM lesions while minimizing the dose to healthy brain tissues; however, in the our opinion, the maximum limit is 10 metastatic brain lesions. The FVMAT is a more suitable technique for treating brain OM because of faster treatment time, minimal toxicity, good patient comfort during the treatment process, and excellent tumor control.

## 5. Conclusions

The findings of our study indicate that FVMAT is a safe, highly comfortable treatment technique that may be used to deliver radiotherapy to patients with multiple brain oligometastases within a significantly shorter treatment time without incurring any serious neurological toxicities. More patient accrual in future retrospective studies is needed to better define its role in patients with brain oligometastasis.

**Author Contributions:** C.-Y.Y.: conceptualization, methodology, data curation, writing—original preparation and review and editing, visualization, supervision, funding acquisition, P.-A.L., F.-H.L. and C.-C.H. software, validation. All authors have read and agreed to the published version of the manuscript.

**Funding:** This research was funded by the Tungs' Taichung Metroharbor Hospital medical research, grant number TTMHH-C1110005.

**Institutional Review Board Statement:** This study was conducted according to the guidelines of the Declaration of Helsinki and was approved by the Institutional Review Board of Tungs' Taichung Metroharbor Hospital (IRB number 110050, approved on 21 October 2021).

**Informed Consent Statement:** Patient consent was waived by the IRB due to the retrospective nature of this study, only the patient's CT and MRI images were used for this study, patient identification including names, case number, and address was removed from images to prevent identification.

**Data Availability Statement:** The data presented in this study are available on request from the corresponding author.

**Conflicts of Interest:** The authors declare no conflict of interest. The funders had no role in the design of the study; in the collection, analyses, or interpretation of data; in the writing of the manuscript, or in the decision to publish the results.

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
