# Peer review of "Fractionated Volumetric Modulated Arc Therapy (FVMAT) for Oligometastatic Brain Tumor"

_onco, doi:10.3390/onco3010004_

Round 1

Reviewer 1 Report (Previous Reviewer 2)

Manuscript significantly improved by the authors

This manuscript is a resubmission of an earlier submission. The following is a list of the peer review reports and author responses from that submission.

Round 1

Reviewer 1 Report

Thank you for allowing me to review this nice work. I put some comments in the PDF that can be used to make the manuscript more appropriate. 

Author Response

Dear Reviewer #1

   I have revised the manuscript to the best of my ability, I hope reviewer #1 will find my revision acceptable. please 

Reviewer 2 Report

The manuscript presents a series of 18 patients with cerebral oligometastases treated with radiotherapy using the VMAT technique.

The authors conclude, based on their results, that the technique is safe and therefore to be recommended in this setting. However, this is a non-prospective study, without a preliminary definition of the sample size (in this regard it would be better to explicitly define it as a retrospective study). Therefore such a clear-cut conclusion does not appear justified. Moreover, it is unclear whether the study was approved by an ethics committee.

Other comments:

"table 4 showed that all patient's ADL were maintained at an acceptable score of < 6 six months after FVMAT treatment. Follow-up MRI 6 months after FVMAT" it would be more useful to know in how many patients there was an improvement or stability or worsening of scores.

"Multileaf collimator (MLC) is a recent development that has made intensity modulated radiation therapy or VMAT treatment a reality" I wouldn't call MLCs "a recent development", having been introduced in the 1980s.

"Its (VMAT) use is recommended for clinical practice in the management of brain oligometastases in a community hospital setting with limited resources" I suggest to make this conclusion less assertive, considering the limitations of the study in terms of trial design and sample size .

Author Response

Dear Reviewer #2

    I have revised the manuscript to the best of my ability, I hope reviewer #2 will finid my revision acceptable
